# Size-control in the synthesis of oxo-bridged phosphazane macrocycles via a modular addition approach

Xiaoyan Shi[1,2,5], Felix León[1,5], How Chee Ong[1], Rakesh Ganguly[1,3], Jesús Díaz [4✉] & Felipe García [1✉]

Inorganic macrocycles remain largely underdeveloped compared with their organic coun-terparts due to the challenges involved in their synthesis. Among them, cyclodiphosphazane macrocycles have shown to be promising candidates for supramolecular chemistry applica-tions due to their ability to encapsulate small molecules or ions within their cavities. However, further developments have been handicapped by the lack of synthetic routes to high-order cyclodiphosphazane macrocycles. Moreover, current approaches allow little control over the size of the macrocycles formed. Here we report the synthesis of high-order oxygen-bridged phosphazane macrocycles via a "$3 + n$ cyclisation" ($n = 1$ and 3). Using this method, an all-$P^{III}$ high-order hexameric cyclodiphosphazane macrocycle was isolated, displaying a larger macrocyclic cavity than comparable organic crown-ethers. Our approach demonstrates that increasing building block complexity enables precise control over macrocycle size, which will not only generate future developments in both the phosphazane and main group chemistry but also in the fields of supramolecular chemistry.

[1] Division of Chemistry and Biological Chemistry, School of Physical and Mathematical Sciences, Nanyang Technological University, Singapore, Singapore. [2] School of Materials and Energy, Guangdong University of Technology, Guangzhou, Guangdong, P. R. China. [3] Shiv Nadar University, Greater Noida, India. [4] Departamento de Química Orgánica e Inorgánica, Facultad de Veterinaria Universidad de Extremadura, Cáceres, Spain. [5] These authors contributed equally: Xiaoyan Shi, Felix León. ✉email: jdal@unex.es; fgarcia@ntu.edu.sg

Over the century, organic macrocyclic compounds have been attractive synthetic targets due to their numerous applications in host–guest chemistry, gas storage, and biological systems[1–4]. However, the development of their inorganic counterparts is hindered by their challenging syntheses caused by the low bond energy of element-carbon covalent bonds[5–10]. In this context, cyclodiphosphazane-based macrocycles have drawn attention due to the relatively high bond energy of their saturated P–N bonds, which is comparable with the energy of C–C bond (ca. 290 vs. 348 kJ mol$^{-1}$, respectively)[11].

Throughout past decades, multiple synthetic routes have been developed to obtain purely inorganic phosph(III/III)azane macrocycle frameworks of formula $[P(\mu\text{-}NR)_2(\mu\text{-}X)]_n$ ($n = 2\text{–}5$)[12–16]. Without exception, these pathways have implemented the use of monomeric dichlorocyclodiphosphazane $[ClP(\mu\text{-}NR)]_2$ precursors, and/or its substituted derivatives as building blocks (Fig. 1).

For existing synthetic approaches, it is well-established that macrocycle size is dictated by the nature and steric bulk of bridging atoms or groups present within the macrocyclic backbone (Fig. 1)[13,15,17]. For sterically bulky bridging groups—e.g., N$^i$Pr[14,18,19], N$^t$Bu[15,18–20], or PSiMe$_3$[21]—dimeric macrocycles are favoured (Fig. 1, type B).

Downsizing to bridging NH groups (or O atoms) produces larger tetrameric macrocycles (Fig. 1, type C)[17,22] and, when combined with halide templating (for NH groups), favours pentameric macrocyclic species (Fig. 1, type D)[8,13,22]. However, larger hexameric cyclodiphosph(III/III)azanes macrocycles have been elusive by conventional synthetic routes for the past two decades[8].

Since conventional approaches have reached their limits for the synthesis of high-order cyclodiphosphazane macrocyclic species, as illustrated by the isolation of the first P$^{III}$ pentameric species—representing the largest macrocycle of this kind reported to date—as long ago as 2002[8], we envisaged that the development of more complex building blocks would have the potential to unlock unprecedented macrocyclic species.

Herein, we describe a step-wise strategy for the synthesis of high-order macrocyclic cyclodiphosphazane species using trimeric acyclic O-bridged poly-cyclodiphosphazanes species (poly-P$^{III}_2$N$_2$) as building blocks. Capitalising on these species, we generated—to the best of our knowledge—the first O-bridged hexameric cyclodiphosphazane framework comprising only P$^{III}$ atoms[17,19,23–27]. This report describes the synthesis of trimeric acyclic O-bridged poly-P$^{III}_2$N$_2$ cyclodiphosphazane species and

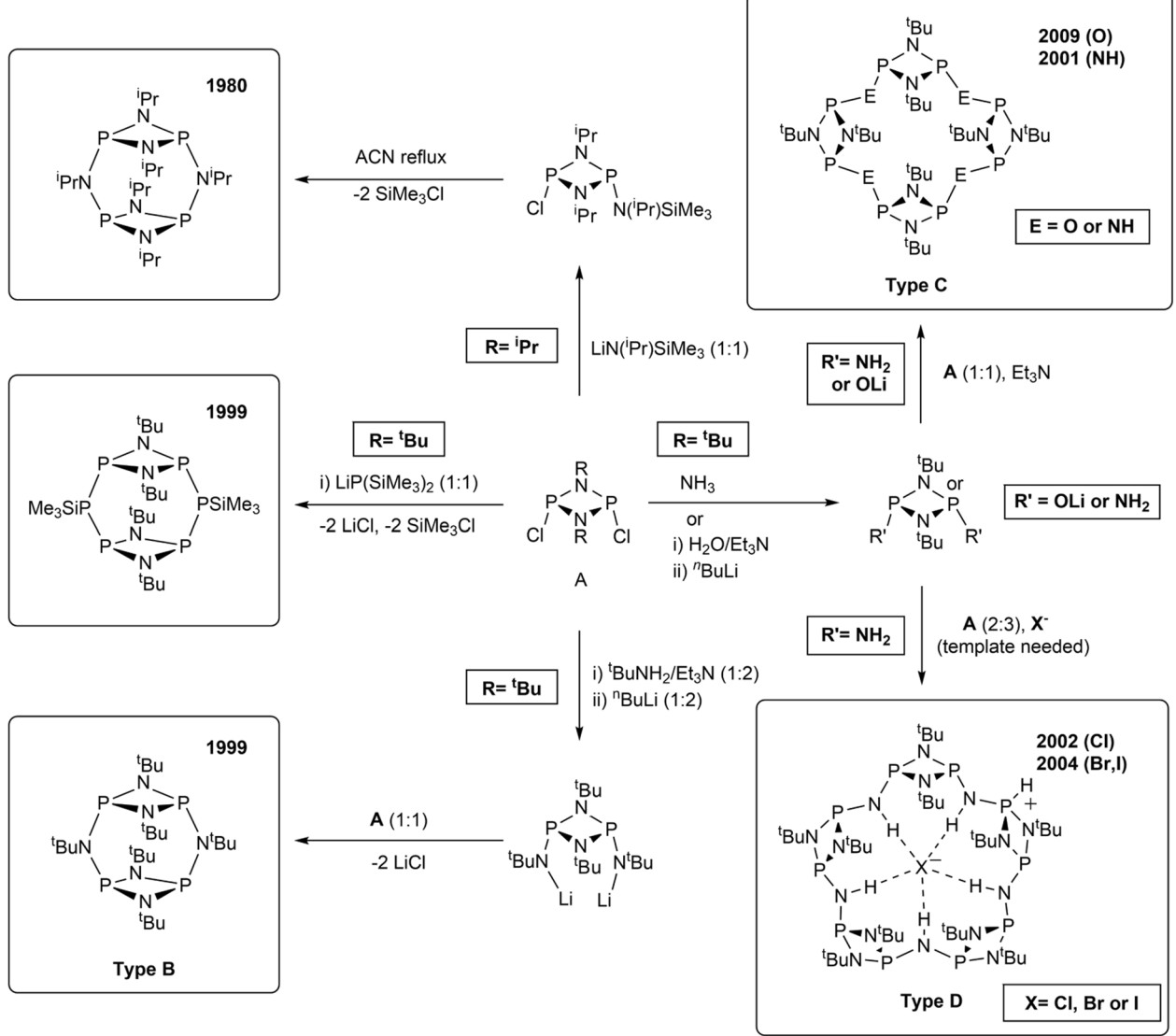

**Fig. 1 Cyclodiphosph(III/II)azanes macrocycles previously reported.** Synthetic routes to all-P$^{III}$ macrocyclic cyclodiphosphazanes dimeric (type B), tetrameric (Type C), and pentameric (Type D) species[13–22].

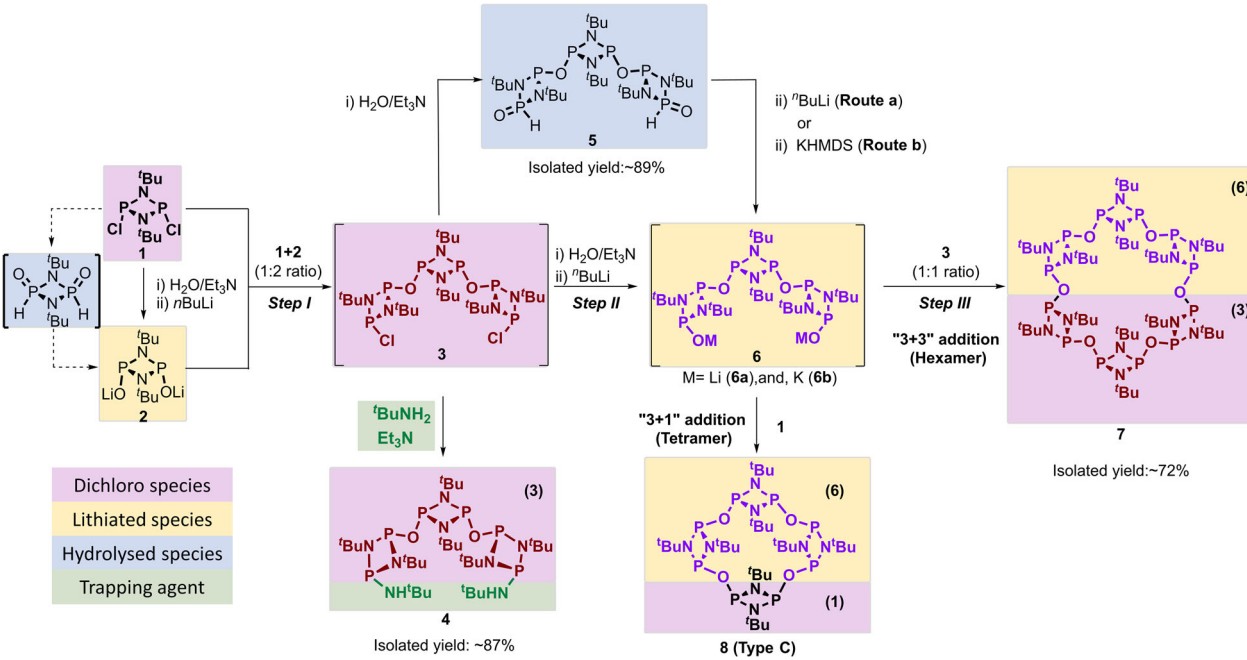

**Fig. 2 Multistep synthetic route to hexameric macrocycle 7.** Detailed synthetic routes to compounds **4**, **5**, and **7**. The different types of building blocks are highlighted in different colours: dichloro (in magenta), metallated (yellow), hydrolysed (blue) and trapping agents (green).

their use as building blocks for the synthesis of a unique hexameric macrocycle—the largest phosph(III/III)azane macrocycle ever reported.

## Results and discussion

**Synthesis of trimeric-$P^{III}_2N_2$ building blocks**. Our studies began with the synthesis of a trimeric acyclic poly-$P^{III}_2N_2$ cyclodiphosphazane *via* a "1 + 2" addition reaction, where 1 mol of [ClP(μ-N$^t$Bu)]$_2$ (**1**), dissolved in THF, was reacted with water in the presence of Et$_3$N, as a Brønsted base, to produce the intermediate [OP (H)(μ-N$^t$Bu)]$_2$, which was subsequently deprotonated with $^n$BuLi to generate [LiOP(μ-N$^t$Bu)]$_2$ (**2**) in situ (see Fig. 2, and Supplementary Scheme S1 and Supplementary Methods). The solution containing **2** was cooled to −78 °C and added dropwise to a THF solution of compound **1** (1:2 ratio) affording the desired trimeric acyclic poly-$P^{III}_2N_2$ {P(μ-N$^t$Bu)}$_2$(μ-O)$_2$[P(μ-N$^t$Bu)$_2$PCl]$_2$ (**3**), comprising two terminal P–Cl moieties for further functionalisation (Fig. 2, Step I). Notably, our procedure allows us to obtain compound **3** in good yields without the formation of undesired macrocyclic species as side products[17].

The formation of compound **3** was monitored by in situ $^{31}$P {$^1$H} NMR, which showed quantitative conversion after 3 h stirring at room temperature. The acyclic trimeric structure **3** displays three resonances in the $^{31}$P{$^1$H} NMR spectrum, a multiplet centred at δ ~198.4 ppm corresponding to the Cl-substituted terminal phosphorus centres. The remaining two phosphorus environments correspond to the two O-substituted phosphorus atoms within the central and two terminal P$_2$N$_2$ rings giving rise to complex multiplets at centred at δ ~162.5 and ~141.4 ppm, respectively (Supplementary Fig. S1). The complex splitting patterns observed are potentially attributed to both a second-order spin system, and the presence of several conformers in solution[28]. Hence, variable temperature (VT) $^{31}$P{$^1$H} NMR studies were performed. Regrettably, our VT $^{31}$P{$^1$H} NMR studies did not show significant changes in the pattern of the signal within the range of temperature studied (i.e., −60 to +50 °C), and hence the complex splitting patterns observed could

not be fully resolved (Supplementary Fig. S2). Compound **3** can be isolated by evaporation of the reaction mixture and filtration in celite (hexanes), to give a yellow waxy solid display $^{31}$P{$^1$H} NMR signals consistent with the ones observed in the in situ spectra. Unfortunately, despite several attempts, crystals suitable for SCXRD studies were not obtained.

To confirm the proposed trimeric nature of the species produced, compound **3** was trapped by reaction with *tert*-butylamine in a 1 to 2 ratio in the presence of excess Et$_3$N (see Supplementary Scheme S2 and Supplementary Methods), to sterically reduce the fluxional behaviour of the multiple conformers observed for **3**—which could not be resolved by VT NMR. Compound **4** exhibits an AB split pattern for the central and medial phosphorus atoms in the $^{31}$P{$^1$H} spectra with two doublets (Δυ/$J_{AB}$ = 1.9)—at δ 140.1 and 139.2 ppm, and a broad singlet 119.2 ppm for the nitrogen substituted terminal phosphorus centres, respectively (Supplementary Fig. S3). The $^1$H NMR spectrum of **4** exhibits one doublet at δ 3.38 which correspond to the terminal NH proton and three *tert*-butyl groups (1:2:1 intensity ratio) on the central and terminal P$_2$N$_2$ rings, and terminal NH$^t$Bu at δ 1.28, 1.50, and 1.56 ppm, respectively (Supplementary Fig. S4). In addition, the $^{13}$C NMR spectrum of **4** presents one triplet at 52.4 ppm and one doublet at 51.7 ppm for the quaternary carbon atoms on *endo*-cyclic *tert*-butyl groups and terminal groups, respectively. Due to the long-distance coupling, the resonances for primary carbon atoms of the *endo*-cyclic *tert*-butyl groups on the central and terminal P$_2$N$_2$ rings appear at 33.2 and 31.8 ppm as a doublet and triplet, respectively. While the secondary carbon atom on the terminal *tert*-butyl group only shows a singlet at 31.9 ppm (Supplementary Fig. S5). Besides, the characteristic stretching for P–N and P–O bonds can be found at 795 and 1011 cm$^{-1}$, respectively, in the IR spectrum of compound **4**. Moreover, diffraction quality crystals of compound **4** were obtained from saturated toluene solution at room temperature. Compounds **4** displays an acyclic poly-cyclodiphosphazane framework comprising three P$_2$N$_2$ rings connected via two oxygen bridging atoms flanked by two NH$^t$Bu moieties (Fig. 3 and Supplementary Fig. S24).

**Closing the circle: synthesis of an hexameric all-$P^{III}_2N_2$ macrocycle via a $3 + 3$ synthetic strategy.** Once the trimeric acyclic nature of **3** was confirmed via the isolation of **4**, we set out to generate a "complementary" trimeric framework with which to attempting the synthesis of high-order macrocycles via "3 + 3" three cyclisation reaction. Hence, in situ generated compound **3** was reacted with two equivalents of water—to form compound {P(μ-N$^t$Bu)}$_2$(μ-O)$_2$[P(μ-N$^t$Bu)$_2$P(=O)H]$_2$ (**5**)—see Supplementary Scheme S3 and Supplementary Methods. The in situ $^{31}$P{$^1$H} spectra of **5** displays three broad signals corresponding to the P(=O)H and P–O atoms in the peripheral $P_2N_2$ rings (at $\delta \sim -3.4$ and ~97.4 ppm, respectively) and the central $P_2N_2$ atoms ($\delta$ 139.3 ppm)—see Supplementary Fig. S6.

To further confirm the trimeric nature of **5**, extraction of the reaction crude of **5** in toluene allowed the isolation of pure **5** in high yields (89%). Compound **5** was characterised using $^{31}$P, and $^1$H spectroscopy (Supplementary Figs. S6–S10), and single-crystal X-ray diffraction studies—crystals were obtained in toluene (−25 °C) overnight. Compound **5** displays an acyclic poly-

cyclodiphosphazane framework comprising peripheral [P(μ-N$^t$Bu)$_2$P(O)H] and central [P(μ-N$^t$Bu)]$_2$ units connected via two oxygen bridging atoms (Fig. 4 and Supplementary Fig. S25). Notably, compounds **4** and **5** represents, to the best of our knowledge, the first isolated examples of oxo-bridged trimeric poly-cyclodiphosphazane $P^{III/III}$ and $P^{III/V}$ species, respectively.

The H-coupled $^{31}$P NMR spectrum of the crystals isolated display a doublet of doublets at $\delta$ −3.4 for the terminal P(O)H with coupling constants of 584 and 14 Hz, $P^V$-H and $P^V$-$P^{III}$, respectively, while both the central $P_2N_2$ and adjacent $P^{III}$-O atoms show broad singlets at $\delta$ 137 and 97 ppm—see Supplementary Figs. S7 and S8. As seen for compound **3**, VT $^{31}$P NMR studies did not show any noticeable changes over the range of temperatures studied (−90 to +90 °C) (Supplementary Fig. S10).

Following the successful isolation of compound **5**, subsequent deprotonation with $^n$BuLi to afford the lithiated acyclic trimeric poly-cyclodiphosphazane {P(μ-N$^t$Bu)}$_2$(μ-O)$_2$[P(μ-N$^t$Bu)$_2$POLi]$_2$ (**6**) (Fig. 2, Step II, and Supplementary Scheme S4). The formation of intermediate **6** was first confirmed by in situ $^{31}$P{$^1$H} NMR (Supplementary Fig. S11) and high-resolution mass spectrometry (HRMS). In addition, the HRMS of **5**, and **6** show the corresponding ion molecular weight [M + 1]$^+$ peaks at 679.2901 and 691.3049 *m/z*, respectively, which is consistent with their proposed structures (Supplementary Figs. S19 and S20, respectively).

In contrast to **5**, the in situ generated compound **6** displayed only two broad resonances in a 1:1 ratio at $\delta$ ~134.7 and 128.6 ppm in its $^{31}$P{$^1$H} NMR spectrum (Supplementary Fig. S11). However, deprotonation of **5** with an alternative base (i.e., NaHMDS) led to the in situ formation of {P(μ-N$^t$Bu)}$_2$(μ-O)$_2$[P(μ-N$^t$Bu)$_2$PONa]$_2$ (**6b**) which displays three signals of equal ratio in the in situ $^{31}$P{$^1$H} NMR with two doublets at ~178.3 and 157.4 ppm, with a $J = 108$ Hz, and a broad singlet at 151.8 ppm—Supplementary Fig. S12. The unexpected $^{31}$P{$^1$H} NMR pattern observed for **6** is attributed to the formation of a lithiated cluster/cage species in solution, a feature previously seen for monomeric counterparts[27].

The ultimate step involved a "3 + 3" cyclisation reaction between compounds **3** and **6** (or **6b**) all generated in situ (Fig. 2, Step III, and Supplementary Scheme S5 and Supplementary Methods). The reaction was monitored by $^{31}$P{$^1$H} NMR showing a quantitative formation of compound **7** after overnight stirring at room temperature. The $^{31}$P{$^1$H} NMR displays one singlet in

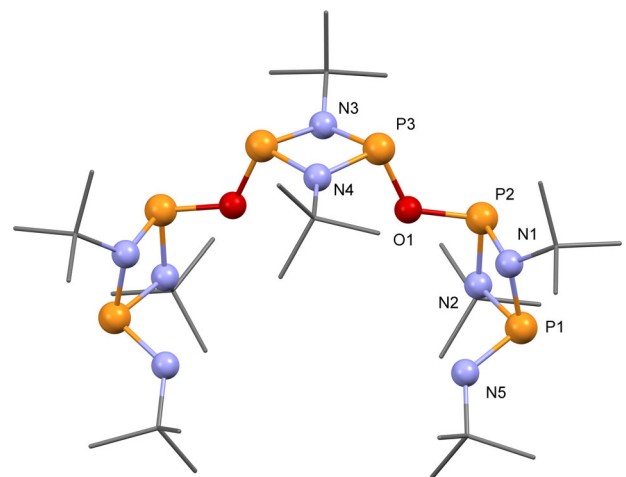

**Fig. 3 Trimeric poly-cyclophosphazane (3) trapped by reaction with $^t$BuNH$_2$.** Solid-state structure of **4**. The tert-butyl groups are drawn as wireframes in all the graphical representations. H atoms are omitted for clarity. For thermal ellipsoid representation, selected bond lengths, and angles see Supplementary Fig. S24.

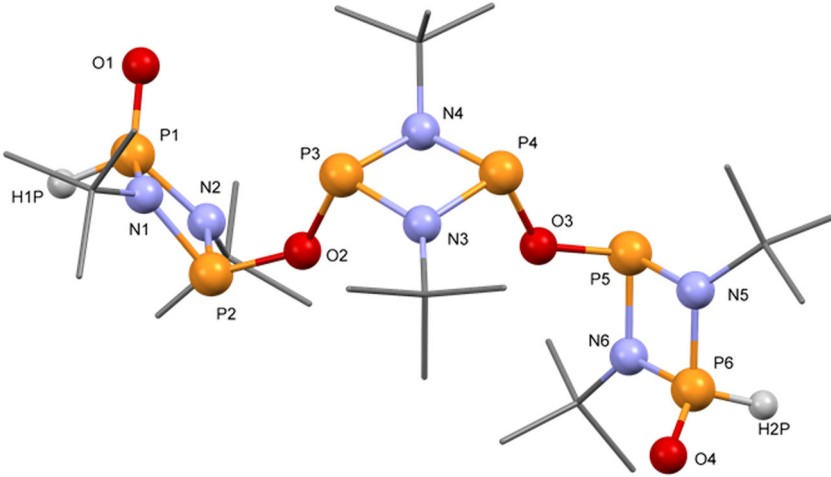

**Fig. 4 Hybrid $P^{III}/P^V$ trimeric poly-cyclophosphazane (5).** Solid-state structure of **5**. The tert-butyl groups are drawn as wireframes in all the graphical representations. H atoms are omitted for clarity. For thermal ellipsoid representation, selected bond lengths, and angles see Supplementary Fig. S25.

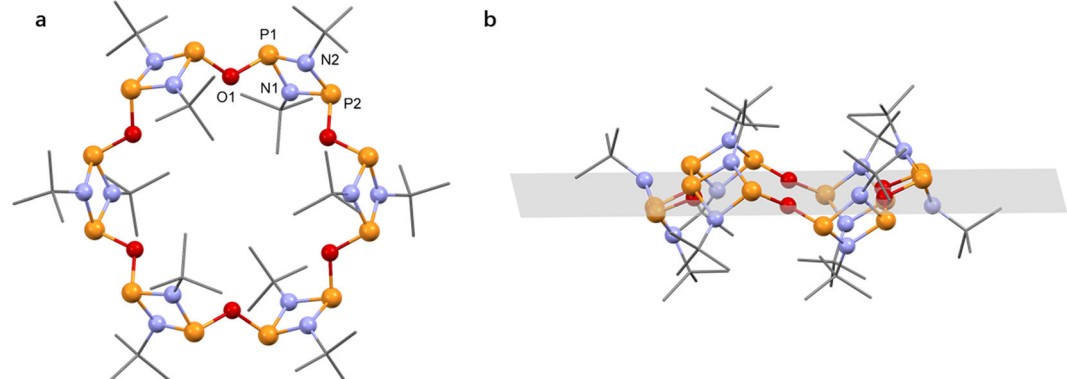

**Fig. 5 Hexameric cyclophosphazane macrocycle (7).** Solid-state structures of **7**. **a** Top view of compound **7**. **b** Side view of compound **7** including the plane defined by all the oxygen bridging atoms. The tert-butyl units are drawn as wireframes in all the graphical representations. H atoms are omitted for clarity. For thermal ellipsoid representation, selected bond lengths, and angles see Supplementary Fig. S26.

the P$^{III}$ region ($\delta$ ~136 ppm), indicating magnetic equivalence of all phosphorus centres present in **7**, suggesting the presence of a highly symmetric backbone (Supplementary Fig. S14). Moreover, the proton NMR spectrum of **7** shows a broad singlet at $\delta = 1.38$ ppm, which is comparable with previously reported oxygen bridged tetrameric macrocyclic species (cf. 1.59 ppm for type C (E=O) in Fig. 1) (Supplementary Fig. S15)[17]. Besides, the $^{13}$C NMR spectrum of presents two singlets at 32.3 and 52.7 ppm for primary carbon atoms and quaternary carbon atoms for all the *endo*-cyclic *tert*-butyl groups due to the high symmetry of structure of compound **7**. Moreover, characteristic stretching bands for P–O bonds can be found at 1066 cm$^{-1}$ in the IR spectrum of compound **7**. Hence, all the spectroscopic analyses are consistent with compound **7** being, to the best of our knowledge, the first all-P$^{III}$ hexameric macrocyclic cyclodiphosphazane $\{(\mu\text{-}O)[P(\mu\text{-}N^tBu)]_2\}_6$ to be isolated.

The cyclic nature of compound **7** was further confirmed by single-crystal X-ray diffraction analysis, from crystals obtained by slow evaporation of the reaction mixture in hexane (Fig. 5a, b). Compound **7** comprises six $P_2N_2$ units connected by bridging oxygen atoms, forming the hexameric macrocyclic cyclodiphosphazane $\{(\mu\text{-}O)[P(\mu\text{-}N^tBu)]_2\}_6$ (**7**), which is consistent with the proposed structure from our spectroscopic analyses.

The mean P–N bonds distance in **7** (i.e., 1.704 Å) is comparable with the standard P–N bond lengths in other cyclodiphosphazane macrocycles[8,10,12,13,17]. Notably, the P–E–P angle decreases from tetrameric (type C) (126° and 129° for E=O and NH, respectively) through pentameric (type D) species (121° for E=NH and X=Cl) to hexamer **7** (118°) which brings neighbouring $P_2N_2$ units closer together. The $P_2N_2$ units present in **7** adopt a puckered zig-zag arrangement, where the $P_2N_2$ rings lean in and out of the main macrocycle plane (Fig. 5b), presumably to reduce steric strain. This is in stark contrast to its smaller tetrameric counterpart (type C), where the $P_2N_2$ rings are all perpendicular to the main macrocycle plane.

**Hexamer cavity properties**. Previous studies on the host–guest chemistry of planar tetrameric crown-ether like phosphazane macrocycles (type C) were unsuccessful due to small-sized cavities[17]. Hence, density functional theory (DFT) ($\omega$-B91 × D/6–31 G(d,p)) calculations were performed for **7**, **18C6** and **21C7** to assess the relative geometry, cavity size and host–guest properties of the new hexamer. The results obtained from both the optimised structures and solid-state structures show a much larger internal area for compound **7** than **21C7** and **18C6** (*ca.* 41, 29 and 22 Å$^2$, respectively), which is also illustrated by their differential interaction with alkali metal cations (Supplementary

Fig. S28). Moreover, the six endocyclic *tert*-butyl groups—located three above and three below the bridging oxygen atoms that delimit the enclosed cavity—define an irregular icosahedron with a calculated internal volume of 120 Å$^3$, which is comparable to that of C60 (see Fig. 6a, b, respectively, and Supplementary Discussion).

Unfortunately, attempts to obtain conclusive experimental evidence for the formation of host–guest adducts between the molecules mentioned above and **7** were unsuccessful. This is attributed to the spatial distribution of the $^tBu$ groups, which somewhat prevent easy access of the guest molecules into the cavity. Currently, we are focusing our efforts in the synthesis of analogous hexameric macrocycles with less sterically hindering substituents to facilitate more efficient host–guest interactions.

Due to the lack of experimental data, and since it has been demonstrated that density functional theory (DFT) studies on the host–guest ability of cyclophosphazane species result in good agreement between theoretical and experimental values[28], we performed DFT calculations to assess the host capacity of **7** relative to the classic crown ethers **18C6** and **21C7** using K$^+$, [NH$_4$]$^+$, [1,1′-biphenyl]-2,2′-diamine, and [1,1′-biphenyl]-2,2′-diammonium as guest molecules (see Supplementary Discussion).

Our studies indicate that in the case of purely inorganic guests (i.e., potassium and ammonium cations), crown ethers display an overall higher binding energies than **7** (i.e., −27.4 and −27.1 vs. −13.7 kcal mol$^{-1}$ for K$^+$, and −29.09 and −29.2 vs. −17.2 kcal mol$^{-1}$ for [NH$_4$]$^+$, with **18C6**, **21C7**, and **7**, respectively. Supplementary Tables S1 and S2). However, in the case of K$^+$ the calculated binding energy for **7** is relatively high considering both the asymmetric nature of the interaction (Supplementary Figs. S31–S33 and Table S1) and the presence of only two strong interactions below 3 Å (cf. six and seven short contacts in **18C6** and **21C7**, respectively). When [NH$_4$]$^+$ is used a guest molecule, both crown ether display three non-bonding hydrogen bonds (HB) (Supplementary Figs. S34a and S35a), whereas only two HB interactions are present in **7** (Supplementary Discussion), which suggests that the relative binding energy of **7** (*ca.* −8.6 kcal mol$^{-1}$ per interaction, Supplementary Table S2) is comparable to those displayed by both **18C6** and **21C7** crown ethers (*ca.* −9.7 kcal mol$^{-1}$). In addition, and in contrast to crown ethers, Electrostatic potential (ESP) analysis show that the cavity of **7** displays the widespread presence of electrostatic potential minima across the cavity (Supplementary Discussion and Figs. S40–S41). Overall, our theoretical studies demonstrate that **7**, despite the asymmetric nature of the host–guest interactions observed, displays relatively high binding

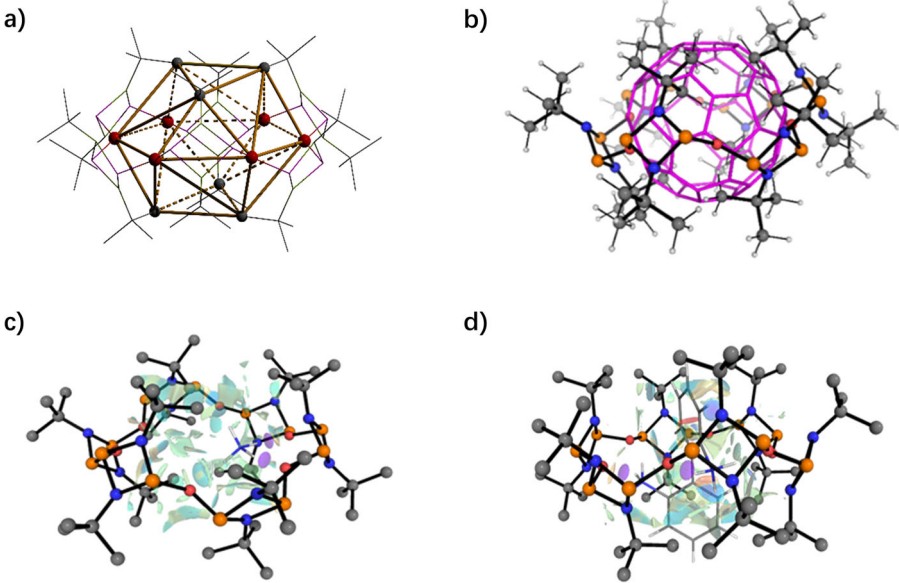

**Fig. 6 Cavity size assessment and non-covalent interactions present in 7.** Side view of the internal volume defined by an icosahedron determined by the oxygen atoms and the *tert*-butyl groups in compound **7** (**a**). Graphical comparison of the cavity sizes of **7** and C60. The geometry optimisation of **7** and C60 was performed at the ω-B91xD/6–31 G(d,p) level of theory. The C60 backbone is represented as wires (**b**). Non-Covalent Interaction (NCI) computed at ω-B91xD/6-31G(d,p) for the host-guest adducts of **7** with then ammonium (**c**) and 1,1′-biphenyl-2,2′-diammonium (**d**).

energies, which combined with the large number of electrostatic potential minima within its cavity, suggest **7** as a more basic host than the studied crown ethers (i.e., **18C6**, **21C7**).

In the case of larger neutral and charged organic molecules (i.e., [1,1′-biphenyl]-2,2′-diamine and [1,1′-biphenyl]-2,2′-diaminium cation), compound **7** displays a more favourable host–guest interaction with both guests (Supplementary Figs. S34b–S36b and Table S3). The differences displayed between the hosts is attributed to both the smaller cavity size present in **18C6**, and **21C7** crown ethers, and the presence of an amphiphilic cavity in **7**. Compound **7** comprises two well-defined zones, two non-polar regions comprising three tert-butyl groups each, sandwiching a central polar region defined by the oxygen atoms along the cavity equator. The amphiphilic nature of the macrocyclic cavity enables hosting large molecules comprising both polar and non-polar regions (such as [1,1′-biphenyl]-2,2′-diaminium). The presence of both polar and non-polar interactions is supported by NCI analyses, which display attractive host–guest interactions in both the polar and non-polar regions present in **7** (Fig. 6c, d, and Supplementary Discussion).

The study of the ESP surface for both crow ethers and **7** provides with further rationalisation of the observed host capacity. While **18C6** and **21C7** show a highly localised negative region within the internal plane defined by the oxygen atoms, **7** displays a much more uneven distribution of the negative charge, as well as positive regions that can stabilise electron poor fragments (Supplementary Fig. S40).

Overall, the large cavity within hexameric macrocycle **7**, combined with the calculate favourable affinities to both organic and inorganic guest molecules offers an exciting opportunity for host-chemistry within the cyclophosphazane arena, which we will be exploring through the development of analogous hexameric macrocycles with less sterically hindering substituents to facilitate more efficient host–guest interactions.

**Accessing other macrocycles via "3 + n" addition reactions.** After the successful synthesis of a high-order hexameric macrocycle, we set out to demonstrate that the use of trimeric species enables control over macrocycle size—which is not accessible using conventional synthetic approaches. As a proof-of-concept, we targeted the synthesis of the oxo-bridged type C tetrameric macrocycle **8** (Scheme 1, type C: E=O), which was previously reported from the reaction of compounds **1** and **2** in a 1:1 ratio—see Supplementary Scheme S6 and Supplementary Methods[17]. Therefore, a "3 + 1" cyclisation between **6** and **1** in a 1:1 ratio was carried out and its progress monitored by $^{31}P\{^1H\}$ NMR. The selective formation of tetramer **8** (see Scheme 2) was evidenced by the appearance of a singlet at δ ~175.0 ppm cf. δ 176.7 ppm for **8**: E=O) and the observation of a peak at 881.3749 (g·mol$^{-1}$, [M + 1]$^+$) in its HRMS spectrum (Supplementary Figs. S16 and S23).

In the case of the "3 + 2" addition, only a handful of dimeric species have been previously described[28–30], among them, the only suitable building block for the selective synthesis of an all P$^{III}$ oxo-bridged pentameric macrocycle [(μ-O){P(μ-N$^t$Bu)}$_2$P(H)O}$_2$][30]. Unfortunately, this compound can only be obtained as mixture with [(μ-N$^t$Bu)$_2$P(H)O]$_2$ in very low yields—which lead to complex mixtures of products that could not be separated. We are currently investing efforts on the development of selective and high-yielding synthetic routes to this dimeric-P$_2$N$_2$ building block, which we hope will grant access to unprecedented pentameric P$^{III}$ oxo-bridged macrocyclic species.

Overall, our results successfully demonstrate the selective formation of hexamers and tetramers by straightforward "3 + 3" and a "3 + 1" cyclisation reactions, respectively. Hence, the herein reported "3 + n" approach provides a synthetic route to enable control over macrocycle size, which has been one of the main synthetic challenges since the macrocyclic phosphazanes were first reported.

In conclusion, we have shown that the newly synthesised acyclic trimeric cyclodiphosphazanes of the type {P(μ-N$^t$Bu)}$_2$(μ-O)$_2$[{P(μ-N$^t$Bu)}$_2$X]$_2$ (where X=Cl or OX −X=Li, Na)) offer a simple and innovative approach to produce high-order Oxo-bridged macrocycles. Most importantly, our strategy allows control over macrocyclic size, which is not possible using established synthetic methodologies. The application of this strategy to achieve families of either high-order or more

sophisticated macrocycles with more accessible cavities is currently under investigation in our group.

Our theoretical studies indicate that, in contrast to crown ethers, compound **7** displays affinity to both inorganic and organic cations due to the presence of basic lone pairs and an amphiphilic cavity. The predicted that the dual nature of these types of oxo-bridged phosphazane macrocycles would provide exciting new opportunities in host–guest and supramolecular chemistry in the future.

Finally, we hope the results herein reported inspire main group chemists to invest their efforts on the development of larger and more sophisticated main group frameworks, which in turn will generate exciting opportunities for future developments not only in the macrocyclic and main group chemistry arenas, but also in the chemical field as a whole.

## Methods
**Synthetic procedures**. See Supplementary Methods and Supplementary Schemes S1–S6.

**Characterisation of compounds**. See Supplementary Figs. S1–S26. For NMR Spectra Supplementary Figs. S1–S16. For FTIR and HRMS spectra see Supplementary Figs. S17–S23. For X-ray analyses see Supplementary Table S1, Supplementary Data 1–3, and Supplementary Figs. S24–S26.

## Data availability
The authors declare that the data supporting the findings of this study are available within the paper and its supplementary information files. The X-ray crystallographic coordinates for structures reported in this Article have been deposited at the Cambridge Crystallographic Data Centre (CCDC), under deposition numbers 1889205, 2025868, and 1889207 for compounds **4**, **5**, and **7**, respectively. These data can be obtained free of charge from The Cambridge Crystallographic Data Centre via www.ccdc.cam.ac.uk/data_request/cif or as Supplementary Data 1–3.

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

## Acknowledgements
FG acknowledge A*STAR AME IRG (A1783c0003, and A2083c0050) and a NTU start-up grant (M4080552) for financial support. F.L. thanks A*STAR for PR fellowship. J.D. thanks COMPUTAEX for granting access to LUSITANIA supercomputing facilities.

## Author contributions
F.G. conceived the research, obtained the funding for the project and jointly supervised the work. X.S. and F.L. designed the experiments and performed the synthetic and characterisation experiments. H.C.O. and R.G. collected the crystal data. F.L. and J.D. performed the theoretical studies. All authors analysed the data and participated in drafting and revising the manuscript.

## Competing interests
The authors declare no competing interests.
