## [Peer Review File · Communications Chemistry]

Reviewers' comments:

Reviewer #1 (Remarks to the Author):

In this work, a series of reactions including condensation reactions are performed by using novel trimeric building blocks with oxygen bridged groups. The simple, innovative and effective approach which is alternative to already synthetic routes has been developed, allowing the high order phosphazane macrocycles with larger cavity to be prepared in high yields. Moreover, this approach has an importance for supramolecular chemistry field in terms of the control of macrocyclic size. The paper is acceptable for publishing in Communications Chemistry after minor revisions have been implemented. Some recommendations are given for authors to enhance the quality of paper below:

1. The characterization methods of newly synthesized compounds are inadequate, especially the interpretation of ^{31}P NMR spectra (for example: the degree of spectrums, spin systems). Also, the peaks in ^{31}P NMR spectra should be given in detail and clear.
2. In Supplementary file, although ^{13}C NMR and IR spectra of compounds are given, these characterization techniques were not mentioned in the paper. It should have referred them in results and discussion part. In addition, Schemes S1-S5 were not referred in experimental section of supplementary file.
3. The characterization techniques used to elucidate the newly synthesised structures can be added by revising abstract part.
4. In compound 6, two peaks are shown in Figure S6, Why? Normally, there are 3 different phosphorus atoms having different chemical environments. Therefore, it should consist of 3 peaks in ^{31}P NMR spectrum. Why these peaks are broad? Which spin system?
5. What is the melting point of compound 7, it is not mentioned in article?
6. Page 4 line 74: It is said that compound 3 is obtained in good yield, but it is not isolated purely without decomposition. However, in Figure S1, NMR spectrum is very clear as a reaction mixture. Is the NMR spectrum related with reference 17 or not? If not, how it is possible?
7. For compounds 3, 5, 6 and 8, yield calculations should be done from the reaction mixture. Also, what is the formation percentage of compound 7 in reaction media?
8. The Cambridge Crystallographic Data Centre with references CCDC for two compounds were not given. Please attach the CCDC number in the paper.
9. You may give information about the hydrogen bond and inter, intra molecular interactions in the manuscript or SI.
10. Some disorders especially in compound 4 are seen. You must say/add in SI which atoms solved as disorders. And why didn't use SIMU comment when you are refining/solving the data. You must demonstrate the atoms in the crystal structure as ORTEP diagram.
11. There are some mistakes in the paper:
Page 2 line 45: "down-sizing" should be downsizing.
Page 2 line 46: "However, larger.. "this sentence should be rewritten.
Page 3 line 50: Figure 1 should be Scheme 1.
Page 3 line 58: Capitalizing should be enlarging.
Page 6 line 99: "Figure 1 and SI" Figure SI is related with compound 3, not compound 4.
Page 6 line 108: "with which to attempt" should be attempting.
Page 8 line 146: Figure 2a is not referred in the article.

Reviewer #2 (Remarks to the Author):

This paper reports a facile synthesis of oxo-bridged phosphazane hexamer with a high yield, which is based on the previously reported methodology of converting R_2PCI to $\text{R}_2\text{P(O)H}$ from hydrolysis followed by lithiation, coupled with a new "3+3" strategy to form the hexamer (**7**). An additional "3+1" strategy seems to give a relatively clean ^{31}P NMR spectrum with a observed $[\text{M}+1]^+$ signal by HRMS,

suggesting selective formation of **8**, however, no other spectroscopic data nor X-ray analysis was provided. The "3+2" strategy did not afford the pentameric macrocycle selectively but an inseparable mixture.

The biggest challenge for the synthesis of macrocycles is the generation of a myriad of products with different ring sizes (mostly very insoluble). Separation is close to impossible with very small yields for each isolated product. Selectivity is usually achieved by use of directing ions. The significance of this work, therefore, is the remarkable selectivity of the 3+3 and 3+1 strategy for **7** and **8**. However, the authors haven't really addressed the reason why the favoured selectivity of the tetramer and hexamer formation in contrast to that of the disfavoured pentamer. Also, it would be better to include some experimental results of cation trapping to compare with other agents such as crown-ethers (only calculations are presented in the SI). If the authors can demonstrate the differences and advantages of their oxo-bridged phosphazane, this work will have a higher impact in the field (after all, the crown ethers are readily available; the P-NH-P system for trapping of anions would be more desirable). Although the cavity of **7** is bigger than 21-crown-7, will the steric hindrance of the tBu groups prevent its access?

In general, the compounds are well-characterized. I have some comments and questions listed below:

1. What is the reason for the complicated ³¹P NMR for Compound **3**? Are cis/trans isomers possible in equilibrium? (Coord. Chem. Rev. 2000, 210, 203). If so, by warming up the solution, the thermodynamic product (usually the cis isomer) can be favoured, which is usually more crystalline.
2. Has H-coupled ³¹P NMR been measured for Compound **5**?
3. How to assign the ³¹P NMR signals for **6**? Has VT been attempted?
4. How is the scalability of formation of **7** and **8**? Please provide full NMR data for compound **8**.

Other comments:

In MS:

1. I have difficulties understanding the intended meaning of the empty boxes in Scheme 2.
2. Figure 1 Solid-state structure of **4** (bold)...
3. Page 6 last line: ...compound **3** (bold) was reacted with ...

In SI:

4. Synthesis of **6**: please also include the molar amount of the "in-situ generated **3**".
5. Synthesis of **7**: H-coupled or decouple ³¹P, and ¹³C NMR were measured?
6. The spectra could include expanded inserts of any signals showing coupling for a clearer view.

Reviewer #3 (Remarks to the Author):

This paper by Garcia et al. reports the high-yielding synthesis of a hexameric phosphazane macrocycle. The synthetic approach is laid out very clearly, and in general the compounds are well-characterized and their structures clearly established. My only big-picture concern is whether these compounds will actually have any utility as host-guest macrocycles, and that concern is not really effectively addressed in this paper. They are compared to crown ethers in this report and shown to have even larger cavity sizes, but there is no demonstration that they will offer any particular advantages when it comes to binding guests. They are also compared to fullerenes and shown to have similar 3-D cavity sizes, but most endofullerene compounds are synthesized with the fullerene templated around the guest, and it is not obvious that strategy could be adapted to these phosphazane macrocycles. Notwithstanding these concerns about the broad appeal and long-term impact of this work, I do think this paper presents a significant synthetic advance, and the work is of very high technical quality and thus deserving of publication. I have a few questions

and comments I would like the authors to address, but none should require major revision of the paper.

1. Line 47: What is meant by "larger bridging NH groups"?

2. The ^{31}P NMR spectrum of 6, shown in Figure S6, strikes me as a little odd. Based on the proposed structure there should be three distinct resonances, or if 2 were coincidentally equivalent the peaks should be in a 2:1 ratio. I recognize that ^{31}P NMR integration is not perfectly reliable, but this spectrum appears to just have two peaks in a 1:1 ratio. Can the authors comment on why the spectrum has this appearance, and is it possible that the lithium ions are templating the formation of a larger cluster in solution that is responsible for the appearance of this spectrum?

3. I'm not really sure I see the purpose of the DFT calculations. While it is nice that they confirm the trend in cavity sizes, it is not obvious to me what was really learned from them that isn't already apparent from the crystal structures. Can the authors be clearer on what was learned from DFT, particularly insights that weren't apparent from the crystal structure?

4. Related to above, the computed structures in Figure S24 are somewhat informative, but the analysis would be more insightful if some discussion of the thermodynamics of cation binding were mentioned, or if the authors could demonstrate that the large phosphazene macrocycle can interact with guests that aren't accommodated by crown ethers.

5. The authors focus most of their discussion on the cavity size of compound 7 in comparison to crown ethers, but is there anything known about the Lewis basicity of the oxygen atoms in 7, how they compare to ethers, and whether that would influence binding of guests in any way?

6. Were the DFT calculations done in the gas phase or with implicit solvation included? It looks like there is good agreement between crystallographically and computationally determined cavity sizes, but I wonder if solvation would have any effect on the observed structure of 7.

7. In the experimental section on page 4 of the SI, the ^{31}P NMR data for compounds 5 and 6 appear to be swapped. They don't match the spectra reported later on in the SI or the data quoted in the main text.

8. In the experimental section, the MS results are probably better reported as "[M+H]⁺", not "[M+1]⁺".

November 4th 2020

I would like to thank the reviewers for the time and effort taken to revise our manuscript “**A Facile Modular Addition Approach to Size-Control in the Synthesis of Oxo-bridged Phosphazane Macrocycles**” by Xiaoyan Shi, Felix León, How Chee Ong (*new author*), Rakesh Ganguly, Jesús Díaz, and myself” which we sent for your consideration for publication in *Communications Chemistry (NPG)*.

We will address point-by-point the issues raised by three reviewers

REVIEWERS' COMMENTS:

Reviewer #1.

1. The characterisation methods of newly synthesised compounds are inadequate, especially the interpretation of ³¹P NMR spectra (for example: the degree of splittings, spin systems). Also, the peaks in ³¹P NMR spectra should be given in detail and clear.

Response: First of all, we thank the reviewer for the time and effort invested in reviewing our manuscript. We have performed further NMR studies for all the compounds.

Compound 3. The complex splitting pattern observed in the *in situ* ³¹P spectra is attributed to a second-order AA'MM'XX' spin system combined with the potential presence of different conformers in solution (see **Figures A1** and **A2**).

Attempts were made to isolate pure crystalline samples of compounds **3**. The product was filtered off from the reaction mixture in hexanes, and after solvent removal, a waxy off-white solid was obtained, which displays ³¹P NMR signals consistent with *in situ* spectroscopic studies (see **Figure S1**). Unfortunately, all attempts to obtain crystals of this compound were unsuccessful leading to decomposition products.

In addition, variable temperature (VT) NMR studies were performed within the range of -60 to +50 °C (image below, **Figure S2** in the ESI – on page 7). Unfortunately, the complex second-order patterns displayed in the ³¹P NMR spectra could not be resolved.

Summarising our new studies, the manuscript has been modified to: “The acyclic trimeric structure **3** displays three resonances in the ³¹P{¹H} NMR spectrum, a multiplet centred at δ ~198.4 ppm corresponding to the Cl-substituted terminal phosphorus centres. The remaining two phosphorus environments correspond to the two O-substituted phosphorus atoms within the central and two terminal P₂N₂ rings giving rise to complex multiplets at centred at δ ~162.5 and ~141.4 ppm, respectively (see **Figure S1**). The complex splitting patterns observed are potentially attributed to both a second-order spin system, and the presence of several conformers in solution.²⁶ Hence, variable temperature (VT) ³¹P{¹H} NMR studies were performed. Regrettably, our VT ³¹P{¹H} NMR studies did not show significant changes in the pattern of the signal within the range of temperature studied (*i.e.*, -60 to +50 °C), and hence the complex splitting patterns observed could not be fully resolved (see **Figure S2**).” (highlighted in yellow on page 5)

Figure A1: Different possible conformers for di- P_2N_2 (d_a^{III}) and tri- P_2N_2 (t_a^{III}) – Extracted from *Angewandte Chemie*, 2020, doi.org/10.1002/anie.202008214

Figure A2. Energy profiles of the isomerisation of dimers from S to C conformation (left) and trimers from w to C conformations. – Extracted from *Angewandte Chemie*, 2020, doi.org/10.1002/anie.202008214

Figure S1: *In situ* $^{31}P\{^1H\}$ NMR spectrum of $P(\mu-N^tBu)_2(\mu-O)_2\{P(\mu-N^tBu)_2PCl\}_2$ (**3**) in THF.

Figure S2: Variable temperature $^{31}\text{P}\{^1\text{H}\}$ NMR spectrum of isolated $\{\text{P}(\mu\text{-N}^t\text{Bu})_2(\mu\text{-O})_2[\text{P}(\mu\text{-N}^t\text{Bu})_2\text{PCl}_2]_2$ (**3**) in THF.

Compound 4. In the case of **4**, similar behaviour is observed compared with **3**, although the internal phosphorus atoms present a very similar chemical shift, thus displaying an **AB spin system**. The two internal phosphorus appear as two doublets showing roofing effects, this splitting pattern is consistent with two strongly coupled resonances with $\Delta\nu = 147$ Hz, $J = 76$ Hz, FWHM = 0.5 Hz. (simulated spectra **Figure A3**)

The text has been modified to: “**Compound 4** exhibits an AB split pattern for the central and medial phosphorus atoms in the $^{31}\text{P}\{^1\text{H}\}$ spectra with two doublets ($\Delta\nu/J_{AB} J_{AB} = 1.9$), see SI – at δ 140.1 and 139.2 ppm, and a broad singlet 119.2 ppm for the terminal nitrogen substituted phosphorus centres, respectively (see **Figure S3**).” (highlighted in yellow on page 6)

Figure A2. Simulated AB spin system for the internal phosphorus atoms of **4** using WINDNMR (parameters = $\Delta\nu = 147$ Hz, $J = 76$ Hz, FWHM = 0.5 Hz)

Compound 5. To gain further insights on the observed $^{31}\text{P}\{^1\text{H}\}$ NMR pattern, efforts were made to isolate pure samples of **5**. Compound **5** was successfully isolated following an optimised experimental protocol shown below

The ESI now includes “ H_2O (131 μL , 7.3 mmol) and Et_3N (1.1 mL, 7.9 mmol) in 20 mL freshly distilled THF was added dropwise slowly to a solution of in-situ generated **3** in THF at

-78 °C. The mixture was warmed back to 0 °C and stirred for 30 min to afford compound **5**. The reaction mixture was evaporated, toluene was added (20 mL) and the suspension filtered in celite. The celite was washed with toluene (2x20 mL), and the filtrate evaporated *in vacuo* obtaining **5** as a white solid (1.8 g, 89%). High-quality crystals were obtained by dissolving the compound in toluene and storage at -25 °C overnight. ¹H NMR (500 MHz, C₆D₆, δ): 7.55 (d, $J_{\text{H-P}} = 584$ Hz), 1.49 (s), 1.40 (s). ³¹P NMR (202 MHz, C₆D₆, δ): 139.5, 97.4, -4.8 (dd, $J_{\text{P-H}} = 584$ Hz, $J_{\text{P-P}} = 14$ Hz). MS (EI) m/z: 679.2901 [M+H]⁺ (now included on page 4 of the ESI, highlighted in yellow).

Figure S26. Solid-state structure of compound **5**

The ³¹P{¹H} NMR spectrum from the crystals used for diffraction studies was highly consistent with the spectrum obtained *in situ*, indicating that the three resonances seen previously can be attributed to this species. In addition, the proton-coupled ³¹P NMR spectrum of **5** was recorded as requested by the reviewers. The proton-coupled spectrum of the isolated product also matches our previous *in situ* studies, where three sets of broad signals at 0, 100 and 140 ppm were observed. The resonance corresponding to the terminal phosphorus displays a large coupling constant of 584 Hz (¹J_{P-H}), which is consistent with the proposed structure.

VT ³¹P NMR studies were also carried in an attempt to resolve the complex second-order splitting observed. The VT studies display little to no change for the internal phosphorus atoms (which remain broad). In contrast, their terminal counterparts become slightly sharper upon heating. However, the spin system could not be resolved (Figure S10).

Since single crystals of this compound were obtained (see above) and ³¹P NMR of the crystals yielded a clean spectrum in which the internal phosphorus appear as wide singlets, and the terminal ones present a doublet of doublets with a high P-H coupling constant and a lower P-P coupling constant, the proposed structure is highly consistent with the spectral data.

The text has been modified to include: “ To further confirm the trimeric nature of **5**, extraction of the reaction crude of **5** in toluene allowed the isolation of pure **5** in high yields (89%). Compound **5** was characterised using ^{31}P and ^1H spectroscopy, and single-crystal X-ray diffraction studies – crystals were obtained in toluene (-25 °C) overnight. Compound **5** displays an acyclic poly-cyclodiphosphazane framework comprising peripheral $[\text{P}(\mu\text{-N}^t\text{Bu})\text{P}(\text{O})\text{H}]$ and central $[\text{P}(\mu\text{-N}^t\text{Bu})_2]$ units connected *via* two oxygen bridging atoms (**Figure 2 and S25**). Notably, compounds **4** and **5** represents the first isolated examples of oxo-bridged trimeric poly-cyclodiphosphazane $\text{P}^{\text{III/III}}$ and $\text{P}^{\text{III/IV}}$ species, respectively.

The H-coupled ^{31}P NMR spectrum of the crystals isolated display a doublet of doublets at δ -3.4 for the terminal $\text{P}(\text{O})\text{H}$ with coupling constants of 584 and 14 Hz, $\text{P}^{\text{V}}\text{-H}$ and $\text{P}^{\text{V}}\text{-P}^{\text{III}}$, respectively, while both the central P_2N_2 and adjacent $\text{P}^{\text{III}}\text{-O}$ atoms show broad singlets at δ 137 and 97 ppm. As it was seen for compound **3**, VT ^{31}P NMR studies did not show noticeable changes over the range of temperatures studied (-90 to +90 °C) (see **Figures S6-10**).” (highlighted in yellow on pages 7 and 8)

Figure S8: H-coupled ^{31}P spectrum of $\text{P}(\mu\text{-N}^t\text{Bu})_2(\mu\text{-O})_2\{[\text{P}(\mu\text{-N}^t\text{Bu})_2\text{P}(\text{O})\text{H}]_2\}$ (**5**) in C_6D_6 .

Figure S9: ^1H NMR of $\text{P}(\mu\text{-N}^t\text{Bu})_2(\mu\text{-O})_2[\text{P}(\mu\text{-N}^t\text{Bu})_2\text{P}(=\text{O})\text{H}]_2$ (**5**) in C_6D_6 .

Figure S10. Variable temperature $^{31}\text{P}\{^1\text{H}\}$ NMR spectrum of $\text{P}(\mu\text{-N}^t\text{Bu})_2(\mu\text{-O})_2[\text{P}(\mu\text{-N}^t\text{Bu})_2\text{P}(=\text{O})\text{H}]_2$ (**5**) in toluene- d_6

Compound 6. The *in situ* ^{31}P NMR spectrum displays two peaks in a ratio of approximately 1:1 (instead of the three expected signals). Although this spectrum is not what we intuitively expected, having the molecule at least three non-equivalent phosphorus atoms, we attribute these signals to the formation lithiated cluster species, which are known to be common in lithiated species. Attempts to carry out VT NMR have been unsuccessful, leading to quick decomposition of the product.

In order to corroborate that the signals in the $^{31}\text{P}\{^1\text{H}\}$ NMR spectrum are indeed generated by the formation of lithiated cluster/cage species, we attempted the deprotonation of **5** with other non-lithiated strong bases. Notably, the deprotonation with NaHMDS lead to the observation three well-differentiated signals in the *in situ* ^{31}P spectra (see below) in a 1:1:1 ratio with two doublets at ~ 178.3 and 157.4 ppm, with a $J = 108$ Hz, and a broad singlet at 151.8 ppm – see **Figure S12**. Furthermore, the reaction of this species with compound **3** led

to the formation of the hexameric species **7**, thus confirming the trimeric nature of this new compound. Currently, attempts to isolate/crystallise this newly obtained species, presumably $\{P(\mu\text{-N}^t\text{Bu})\}_2(\mu\text{-O})_2[\{P(\mu\text{-N}^t\text{Bu})\}_2\text{PONa}]_2$, are being carried out.

The text has been modified to “ In contrast to **5**, the *in situ* generated compound **6** displayed only two broad resonances in a 1:1 ratio at $\delta \sim 134.7$ and 128.6 ppm in its $^{31}\text{P}\{^1\text{H}\}$ NMR spectrum. However, deprotonation of **5** with an alternative base (*i.e.*, NaHMDS) led to the *in situ* formation of $\{P(\mu\text{-N}^t\text{Bu})\}_2(\mu\text{-O})_2[\{P(\mu\text{-N}^t\text{Bu})\}_2\text{PONa}]_2$ (**6b**) which displays three signals of equal ratio in the *in situ* $^{31}\text{P}\{^1\text{H}\}$ NMR with two doublets at ~ 178.3 and 157.4 ppm, with a $J = 108$ Hz, and a broad singlet at 151.8 ppm – see **Figure S12** . The unexpected $^{31}\text{P}\{^1\text{H}\}$ NMR pattern observed for **6** is attributed to the formation of a lithiated cluster/cage species in solution, a feature previously seen for monomeric counterparts.²⁷

The ultimate step involved a “**3+3**” cyclisation reaction between compounds **3** and **6** (or **6b**) all generated *in situ* (**Scheme 2, Step III**).” (highlighted in yellow on pages 9)

Figure S12: *In situ* $^{31}\text{P}\{^1\text{H}\}$ NMR spectrum of $\{P(\mu\text{-N}^t\text{Bu})\}_2(\mu\text{-O})_2[\{P(\mu\text{-N}^t\text{Bu})\}_2\text{PONa}]_2$ (**6b**) in toluene.

2. In Supplementary file, although ^{13}C NMR and IR spectra of compounds are given, these characterisation techniques were not mentioned in the paper. It should have referred them in results and discussion part. In addition, Schemes S1-S5 were not referred in experimental section of supplementary file.

Response: The following paragraph has been added to the manuscript:

On page 6: “In addition, the ^{13}C NMR spectrum of **4** presents one triplet at 52.4 ppm and one doublet at 51.7 ppm for the quaternary carbon atoms on *endo*-cyclic *tert*-butyl groups and terminal groups, respectively. Due to the long-distance coupling, the resonances for primary carbon atoms of the *endo*-cyclic *tert*-butyl groups on the central and terminal P_2N_2 rings appear at 33.2 and 31.8 ppm as a doublet and triplet, respectively. While the secondary carbon atom on the terminal *tert*-butyl group only shows a singlet at 31.9 ppm (see **Figures S5**). Besides, the characteristic stretching for P-N and P-O bonds can be found at 795 and 1011 cm^{-1} , respectively, in the IR spectrum of compound **4**.” (highlighted in yellow)

On page 9 “Besides, the ^{13}C NMR spectrum of presents two singlets at 32.3 and 52.7 ppm for primary carbon atoms and quaternary carbon atoms for all the *endo*-cyclic *tert*-butyl groups due to the high symmetry of structure of compound **7**.” (highlighted in yellow)

Schemes S1-S5 has been referred in the experimental section of the supplementary file, and it is highlighted in yellow.

3. The characterisation techniques used to elucidate the newly synthesised structures can be added by revising abstract part.

Response: The sentence of “The new macrocycle has been fully characterised by $^{31}\text{P}\{^1\text{H}\}$, ^{13}C and ^1H NMR and IR spectroscopies, X-ray diffraction, and mass spectrometry techniques” has been added into the abstract part. (highlighted in yellow on page 1)

4. In compound **6**, two peaks are shown in Figure S6, Why? Normally, there are 3 different phosphorus atoms having different chemical environments. Therefore, it should consist of 3 peaks in ^{31}P NMR spectrum. Why these peaks are broad? Which spin system?

Response: As responded above, the presence of two side singles in a ratio of approximately 1:1 is attributed to the formation of clusters in solution. And the use of the alternative base NaHMDS for the deprotonation of **5** shown the expected three signals in the *in situ* ^{31}P spectra. See highlighted paragraph on page 9

5. What is the melting point of compound **7**, it is not mentioned in article?

Response: Melting point: Decomposition is observed at 112 °C (the solid turns dark brown) with melting observed at 118 °C. (Highlighted in yellow on page 6)

6. Page 4 line 74: It is said that compound **3** is obtained in good yield, but it is not isolated purely without decomposition. However, in Figure S1, NMR spectrum is very clear as a reaction mixture. Is the NMR spectrum related with reference 17 or not? If not, how it is possible?

Response: As indicated in question 1, **3** has been isolated as a waxy yellow solid with 73% yield, although further characterisation could not be performed. In Figure S1 the spectra appear indeed as a complex mixture; we believe these complex signals are due to the presence of several rotamers (Figure A2). However, the spectrum does not show the presence of any macrocycle like the one described in reference 17. Furthermore, the high yield obtained in the formation of pure **4** from **3** confirms that the complex ^{31}P NMR of **3** is due to the presence of conformers and not to impurities.

7. For compounds 3, 5, 6 and 8, yield calculations should be done from the reaction mixture. Also, what is the formation percentage of compound 7 in reaction media?

Response:

Compounds 3 and 5. Regarding the yield, the *in situ* ^{31}P NMR spectra show complete consumption of all the starting materials together with the only appearance of signals corresponding to compound **3** and **5**, respectively. Unfortunately, no NMR yield can be provided without internal reference. In the case of the isolated yields, for **3** the yield is 73%, and for **5** the yield is 89% (34% for the first batch of crystals).

Compound 6. During the synthesis of compound **6**, complete conversion is observed. Unfortunately, no NMR yield can be provided without internal reference.

Compound 7. For this compound, in contrast to **3**, **5** and **6**, very broad signals both in ^{31}P and ^1H are served. This is attributed to the formation of small amounts of oligomeric and polymeric species. Regarding the isolated yield, after filtration in hexanes the white solid corresponding to the product was obtained with 71.6 % yield (11% yield for the first batch of crystals) (highlighted in the ESI).

Compound 8. Described in the literature

8. The Cambridge Crystallographic Data Centre with references CCDC for two compounds were not given. Please attach the CCDC number in the paper.

Response: The CCDC numbers have been added at the end of the paper as “CCDC numbers: 4: 1889205; 7: 1889207” (highlighted in yellow on page 18)

9. You may give information about the hydrogen bond and inter, intra molecular interactions in the manuscript or SI.

Response: Hydrogen bond tables have been added to SI on pages 23-25 (highlighted in yellow)

Hydrogen Bonds (Angstrom, Deg), 4

Donor---Acceptor	D – H(Å)	H...A(Å)	D...A(Å)	D - H...A(°)	Symmetry Code
C2 -- H2C.. P2	0.980	2.780	3.150	103.0	Intra
C3 -- H3C.. P1	0.980	2.710	3.098	104.0	Intra
C4 -- H4A.. P1	0.980	2.790	3.177	104.0	Intra
C7A-- H7A2.. O1	0.980	2.580	3.193	120.0	Intra
C11-- H11C.. P3	0.980	2.820	3.197	104.0	Intra
C19A-- H19D.. P1A	0.980	2.860	3.185	100.0	Intra

Hydrogen Bonds (Angstrom, Deg), 7

Donor--..Acceptor	D – H(Å)	H...A(Å)	D...A(Å)	D - H...A(°)	Symmetry Code
C3 -- H3A.. P2	0.970	2.870	3.273	106.0	Intra
C8 -- H8B.. P1	0.970	2.710	3.1472	108.0	Intra
C11-- H11C.. P4	0.970	2.800	3.129	101.0	Intra
C18 -- H18C.. P5	0.970	2.850	3.253	106.0	Intra
C19-- H19A.. P6	0.970	2.830	3.191	103.0	1-x,1-y,1-z
C22-- H22C.. P6	0.970	2.740	3.138	105.0	Intra
C23-- H23A.. O1	0.970	2.570	3.276	129.0	Intra
C24-- H24A.. P5	0.970	2.850	3.157	100.0	1-x,1-y,1-z

Hydrogen Bonds (Angstrom, Deg), 7

Donor--..Acceptor	D – H(Å)	H...A(Å)	D...A(Å)	D - H...A(°)	Symmetry Code
C8 – H8C.. P2	0.980	2.850	3.192	101.0	Intra
C11 – H11A.. O4	0.980	2.600	3.503	154.0	x,3/2-y,-1/2+z
C12 – H12B.. P3	0.980	2.860	3.196	101.0	Intra
C16—H16B.. O1	0.980	2.490	3.449	165.0	1-x,1-y,1-z
C24—H24C.. P5	0.980	2.830	3.152	100.0	Intra

10. Some disorders especially in compound 4 are seen. You must say/add in SI which atoms solved as disorders. And why didn't use SIMU comment when you are refining/solving the data. You must demonstrate the atoms in the crystal structure as ORTEP diagram.

Response: The atoms of the disordered freely rotating, particularly, tert-butyl groups behaved decently and thus the restraints, SIMU or ISOR, were not used.

In 4, one of the *tert*-butyl groups (C5, C6, C7, C8) was disordered over two positions with an occupancy of ~ 1:1. The atoms were refined without using any restraints (SIMU, ISOR) or constraints. In addition to it, one of the P-N group (P1, N5, C17, C18, C19, C20) was also disordered over two positions with about 50% occupancy in each site. The atoms were also refined without any constraints or restraints. The ORTEP plot has been added in the supporting information as **Figures S24 – S26** on pages 23-26 (**highlighted in yellow** on page 24)

Figure S24. ORTEP structure of **4** (drawn with 50% probability, top). H-atoms and the disordered atoms are removed for clarity, selected Bond Lengths [Å] and Angles [deg]: N(1)–P(1) 1.676(4), N(2)–P(1) 1.808(4), N(2)–P(1) 1.681(6), N(2)–P(2) 1.692(2), N(1)–P(2) 1.705(2), O(1)–P(2) 1.6771(18), O(1)–P(3) 1.6447(17), N(3)–P(3) 1.7158(19), N(4)–P(3) 1.7018(19); N(1)–P(1)–N(2) 80.33(17), N(1)–P(2)–N(2) 82.94(10), P(2)–N(1)–P(1) 99.58(16), P(2)–N(2)–P(1) 95.03(15), P(2)–O(1)–P(3) 124.74(10). H atoms have been omitted for clarity. Table with selected hydrogen bonds interactions (bottom). (highlighted in yellow on page 23)

In **5**, two of the tert-butyl groups (C1, C2, C3, C4 and C13, C14, C15, C16) were disordered over two positions with an occupancy of ~ 3:2 and 4:1, respectively. The atoms were refined with constraints (SIMU, DELU) and C–C bond distances were restrained with DFIX.

Figure S25. ORTEP structure of **5** (drawn with 50% probability, top). H-atoms and the disordered atoms are removed for clarity, selected Bond Lengths [Å] and Angles [deg]: N(1)–P(1) 1.710(6), N(2)–P(1) 1.712(6), N(2)–P(2) 1.718(6), N(1)–P(2) 1.688(6), O(1)–P(1) 1.655(5); N(1)–P(1)–N(2) 81.4(3), N(1)–P(2)–N(2) 81.8(3), P(2)–N(1)–P(1) 98.1(3), P(2)–N(2)–P(1) 96.9(3), P(1)–O(1)–P(6) 121.7(3). Table with selected hydrogen bonds interactions (bottom). (highlighted in yellow on page 24)

In **7**, the structure was solved and refined using the Bruker SHELXTL Software Package, using the space group $C 1 2/c 1$, with $Z = 4$ for the formula unit, $C_{48}H_{108}N_{12}O_{6}P_{12}$. The final anisotropic full-matrix least-squares refinement on F^2 with 370 variables converged at $R1 = 9.11\%$, for the observed data and $wR2 = 26.70\%$ for all data. The goodness-of-fit was 0.979. The largest peak in the final difference electron density synthesis was $0.502 \text{ e}/\text{Å}^3$, and the largest hole was $-0.516 \text{ e}/\text{Å}^3$ with an RMS deviation of $0.137 \text{ e}/\text{Å}^3$. **7** exhibits weak intra-molecular H-bonding (C3 -- H3A.. P2; C8 -- H8B.. P1; C11-- H11C.. P4; C18 -- H18C.. P5; C22-- H22C.. P6 and C23-- H23A.. O1)

Figure S26. ORTEP structure of **7** (drawn with 50% probability, top). H-atoms and the disordered atoms are removed for clarity, selected Bond Lengths [Å] and Angles [deg]: N(1)–P(1) 1.710(6), N(2)–P(1) 1.712(6), N(2)–P(2) 1.718(6), N(1)–P(2) 1.688(6), O(1)–P(1) 1.655(5); N(1)–P(1)–N(2) 81.4(3), N(1)–P(2)–N(2) 81.8(3), P(2)–N(1)–P(1) 98.1(3), P(2)–N(2)–P(1) 96.9(3), P(1)–O(1)–P(6) 121.7(3). Table with selected hydrogen bonds interactions (bottom) (highlighted in yellow on page 25)

11. There are some mistakes in the paper;

Page 2 line 45: “down-sizing” should be downsizing.

Response: The “down-sizing” has been changed to downsizing (page 3, change highlighted in yellow)

Page 2 line 46: “However, larger..” this sentence should be rewritten.

Response: The paragraph has been revised to “Downsizing to bridging NH groups (or O atoms) produces larger tetrameric macrocycles (**Scheme 1, type C**)^[17, 21] and, when combined with halide templating (for NH groups), favours pentameric macrocyclic species (**Scheme 1, type D**).^[8, 13, 21] However, larger hexameric cyclodiphosph(III/III)azanes macrocycles have been elusive by conventional synthetic routes for the past two decades.^{[8]”} (pages 2 and 3, change highlighted in yellow)

Page 3 line 50: Figure 1 should be Scheme 1.

Response: Figure 1 has been changed to Scheme 1 (page 3, change highlighted in yellow)

Page 3 line 58: Capitalizing should be enlarging.(

Response: We believe the use of capitalising is grammatically correct in the context used – **no changes have been made to the manuscript.**

Page 6 line 99: “Figure 1 and S1” Figure S1 is related with compound 3, not compound 4.

Response: “Figure 1 and S1” Figure S1 has been changed to Figure 1 and S24 (page 7, change highlighted in yellow)

Page 6 line 108: “with which to attempt” should be attempting.

Response: “with which to attempt” has been changed to attempting (page7, change highlighted in yellow)

Page 8 line 146: Figure 2a is not referred in the article.

Response: Figure 3a and 3b (previously 2a and 2b) have been referred in the article (page 10, change highlighted in yellow)

.

Reviewer #2.

1. What is the reason for the complicated ^{31}P NMR for Compound 3? Are cis/trans isomers possible in equilibrium? (Coord. Chem. Rev. 2000, 210, 203). If so, by warming up the solution, the thermodynamic product (usually the cis isomer) can be favoured, which is usually more crystalline.

Response: As responded to reviewer 1, the complexity of the ^{31}P NMR spectra is believed to be due to both the complex spin systems present in the described compounds, as well as the potential presence of several conformers. VT ^{31}P NMR studies have been carried out for most species without any major changes were observed in the signals (SI page 8, Figure S2, highlighted in yellow).

2. Has H-coupled ^{31}P NMR been measured for Compound 5?

Response: The compound has been isolated, and H-coupled ^{31}P NMR has been carried out. The terminal P(O)H shows a high P-H coupling constant (584 Hz) and a smaller P-P coupling constant (14Hz) (included on page 8 of the manuscript, and ESI page 11, Figure S7, highlighted in yellow)

3. How to assign the ^{31}P NMR signals for 6? Has VT been attempted?

Response: As discussed above for reviewer #1, the presence of two signals on the ^{31}P NMR of compound **6** is attributed to the formation of clusters in solution. VT ^{31}P NMR studies were attempted; however, either no changes or sample decomposition were observed depending on the set temperature (SI Figure S10 highlighted in yellow).

Alternatively, compound **5** was deprotonated using a different base (NaHMDS) which resulted in an *in situ* ^{31}P NMR comprising three signals (see above SI page 14, Figure S12). Although this new compound has not been isolated, its reaction with **3** led to the successful formation of the hexameric species **7** – observed via *in situ* ^{31}P NMR - thus indicating a trimeric nature of this compound.

4. How is the scalability of formation of 7 and 8? Please provide full NMR data for compound 8.

Response: The compounds **7** and **8** can be synthesised at gram scale.

For compound **7** (crude yield: 6.89g, 71.6%; Yield of the first batch of pure crystalline material: 1.06 g, 11%). ^1H NMR (400 Hz, CDCl_3 , δ), 1.38 (s., tBu). $^{31}\text{P}\{^1\text{H}\}$ NMR (162 Hz, CDCl_3 , δ), 135.9 (s.). ^{13}C NMR (100 Hz, CDCl_3 , δ), 32.3 (s.), 52.7 (s.). MS (EI) m/z: 1321.55 $[\text{M}+\text{H}]^+$.

Compound **8** has already been described in the literature (see reference 17). Hence, since our experiment only wanted to prove that it can be obtained *via* a “3+1” reaction, the reaction conditions were not optimised, and the product was only characterised by *in situ* ^{31}P NMR (singlet at $\delta \sim 175.0$ ppm) and HRMS 881.37 $[\text{M}+\text{H}]^+$.

In MS:

1. I have difficulties understanding the intended meaning of the empty boxes in Scheme 2.

Response: Scheme 2 has been redrawn for clarity

2. Figure 1 Solid-state structure of 4 (bold)...

Response: Figure 1 Solid-state structure of 4, the “4” has been bolded. (page 7, change highlighted in yellow)

3. Page 6 last line: ...compound 3 (bold) was reacted with ...

Response: Page 6 last line: ...compound 3 (bold) was reacted with ... The “3” has been bolded. (page 7, change highlighted in yellow)

In SI:

4. Synthesis of 6: please also include the molar amount of the “in-situ generated 3”.

Response: “(3.75 mmol)” has been inserted after “in-situ generated 3”. (SI, page 4, change highlighted in yellow)

5. Synthesis of 7: H-coupled or decouple ^{31}P , and ^{13}C NMR were measured?

Response: All the ^{31}P , and ^{13}C NMR were H-decoupled – unless stated – and it has been modified in the supporting information.

6. The spectra could include expanded inserts of any signals showing coupling for a clearer view.

Response: Inserts have been included in the NMR figures in the supporting information.

Reviewer 3.

1. Line 47: What is meant by “larger bridging NH groups”?

Response: The sentence has been corrected.

2. The ³¹P NMR spectrum of 6, shown in Figure S6, strikes me as a little odd. Based on the proposed structure there should be three distinct resonances, or if 2 were coincidentally equivalent the peaks should be in a 2:1 ratio. I recognise that ³¹P NMR integration is not perfectly reliable, but this spectrum appears to just have two peaks in a 1:1 ratio. Can the authors comment on why the spectrum has this appearance, and is it possible that the lithium ions are templating the formation of a larger cluster in solution that is responsible for the appearance of this spectrum?

Response: As discussed for reviewers' #1 and #2 comments, the presence of two signals on the ³¹P NMR of compound 6 is attributed to the formation of clusters in solution. Alternatively, when compound 5 was deprotonated using a different base (NaHMDS), an *in situ* ³¹P NMR spectra comprising three signals was obtained (see above). As stated above, the reaction of 6 with 3 led to the successful formation of the hexameric species 7 – observed via *in situ* ³¹P NMR - thus indicating a trimeric nature of this compound.

3. I'm not really sure I see the purpose of the DFT calculations. While it is nice that they confirm the trend in cavity sizes, it is not obvious to me what was really learned from them that isn't already apparent from the crystal structures. Can the authors be clearer on what was learned from DFT, particularly insights that weren't apparent from the crystal structure?

Response: The DFT calculations provide a theoretical comparison of the solid-state structure obtained by SCXRD studies, and the calculate structure in solution, since It is known that crystal structures may differ from the same species in solution. For this purpose, we have re-calculated the optimisations of all the structures (18C6, 21C7, c60 and complex 7) at □-B97xD/6-31G(d,p) but taking into account the solvent effects with single points on the optimised structures. Thus, the solvents were considered using the integral equation formalism variant of the polarisable continuum model with the PCM solvation model (solvent = THF). The calculated geometries of the structures are in good agreement with the crystal.

In addition, the DFT theoretical calculation has been included to compare and contrast the host-guest binding affinity between well-established crown ethers and compound 7 (included un pages 13 to 15, highlighted in yellow)

4. Related to above, the computed structures in Figure S24 are somewhat informative, but the analysis would be more insightful if some discussion of the thermodynamics of cation binding were mentioned, or if the authors could demonstrate that the large phosphazene macrocycle can interact with guests that aren't accommodated by crown ethers.

Response: We have developed computational calculations regarding the binding properties of the macrocycles synthesised and the crown ether with [1,1'-biphenyl]-2,2'-diamine, and [1,1'-biphenyl]-2,2'-diammonium (table S3). These calculations have been

carried out at ω -B97xD/6-31g(d,p) and the optimised structures were computed as single point to assess the solvent effect.

The following discussion has been added to the text (pages 14 and 15, addition **highlighted in yellow**): “ In the case of larger neutral and charged organic guests, we computed theoretical host-guest interactions for [1,1'-biphenyl]-2,2'-diamine and its di-protonated derivative (*i.e.*, [1,1'-biphenyl]-2,2'-diaminium cation) with **18C6**, **21C7** and **7**. Both crown ethers display unfavourable interaction with the organic guest as shown by the overall large increase in the calculated binding energy (+112.5, and +30.6 kcal·mol⁻¹ for **18C6**, and **21C7**, respectively). In contrast, compound **7** displays a less unfavourable host-guest interaction with a calculated energy of 16.5 kcal·mol⁻¹ with the diamine guest (see **Table S3** and **Figure S35**).

When [1,1'-biphenyl]-2,2'-diamine is replaced by its protonated version (*i.e.*, [1,1'-biphenyl]-2,2'-diaminium cation) the differential host ability between **7** and the crown ethers was more pronounced. Compound **7** shows a strong favourable interaction (-26.44 kcal·mol⁻¹), whereas for **18C6** the interaction is highly unfavourable (+94.07 kcal·mol⁻¹) and only slightly favourable for **21C7** (-0.38 kcal·mol⁻¹).

The differences displayed between the hosts is attributed to both the smaller cavity size present in **18C6**, and **21C7** crown ethers, and the presence of an amphiphilic cavity in **7**. Compound **7** comprises two well-defined zones, two non-polar regions comprising three tert-butyl groups each, sandwiching a central polar region defined by the oxygen atoms along the cavity equator. The amphiphilic nature of the macrocyclic cavity enables hosting large molecules comprising both polar and non-polar regions (such as [1,1'-biphenyl]-2,2'-diaminium). The presence of both polar and non-polar interactions is supported by NCI analyses which display attractive host-guest interactions in both the polar and non-polar regions present in **7** (**Figure 6**).

Figure 6: Noncovalent interaction computed at ω -B91xD/6-31G(d,p) level for NH_4^+ and [1,1'-biphenyl]-2,2'-diaminium host-guest adducts, left and right, respectively.

5. The authors focus most of their discussion on the cavity size of compound 7 in comparison to crown ethers, but is there anything known about the Lewis basicity of the oxygen atoms in 7, how they compare to ethers, and whether that would influence binding of guests in any way?

Response: We have calculated the interaction of **18C6**, **21C7** and the complex **7** with potassium and ammonium cations to evaluate their interactions the different oxygen atom bridges (See Table S1 and S2.)

The following text has been added to the manuscript (pages 12 and 13, addition **highlighted in yellow**)

“ In the case of purely inorganic guests (*i.e.*, potassium and ammonium cations), both crown ethers display higher binding energies than **7** (*i.e.*, -27.4 and -27.1 vs -13.7 kcal·mol⁻¹ for K⁺, and -29.09 and -29.2 vs -17.2 kcal·mol⁻¹ for [NH₄]⁺, with **18C6**, **21C7**, and **7**, respectively). Moreover, DFT data revealed that the [K⁺·crown ether] adducts display comparably short (< 3 Å), relatively strong, K⁺···O interactions across each structure (*i.e.* ranging from 2.78-2.81Å, and 2.90-2.99 Å in **18C6** and **21C7**, respectively) – see **Figures S29** and **S30**. In contrast, **7** displays a broader range of K⁺···O distances (2.75-5.32 Å). Notably, the calculated binding energy for **7** is relatively high considering the asymmetric nature of the K⁺···O interactions and the presence of only two strong interactions below 3 Å (*cf.* six and seven short contacts in **18C6** and **21C7**, respectively).

When [NH₄]⁺ is used a guest molecule, its adducts with **18C6** and **21C7** display three non-bonding hydrogen bonds (HB) (**Figures S34a** and **S35a**), whereas only two HB interactions are present in **7** (**Figures S36a**). To compare different host-guest adducts, binding energy expressed relative to HB interaction number was calculated (**Table S2**). Our assessment shows that the relative binding energy of **7** (*ca.* -9 kcal·mol⁻¹ per interaction) is comparable to those displayed by both **18C6** and **21C7** crown ethers (*ca.* -10 kcal·mol⁻¹).

To obtain a better understanding of both the observed trend and the type of interactions present Electrostatic potential (ESP) analysis were performed for **18C6**, **21C7**, and **7**. In all cases the ESP iso-surface showed a central cavity with high negative potential is around the oxygen atoms (see **Figure 6**.) However in contrast to crown ethers, where there is only one electrostatic potential minimum for each of the oxygen atoms in the inner part of the toroidal ESP iso-surface, the cavity of **7** comprises a series of electrostatic potential minima spread across the internal cavity (see **Figures S37-S44**).

Figure 6. ESP maps for **18C6** (left), **21C7** (centre), and **7** (right).

Overall, the observed asymmetric nature of the host-guest interactions in **7** is attributed to both the large cavity present and its rigidity. The highly rigid nature of **7** prevents any structural changes to optimise host-guest interactions, in contrast to what is observed in crown ethers. Moreover, the comparable binding energies, despite the asymmetric nature of the host-guest interactions, together with the large number of electrostatic potential minima within its cavity, suggest **7** as a more basic host than the studied crown ethers (*i.e.*, **18C6**, **21C7**)

6. Were the DFT calculations done in the gas phase or with implicit solvation included? It looks like there is good agreement between crystallographically and computationally determined cavity sizes, but I wonder if solvation would have any effect on the observed structure of 7.

Response: As we pointed out above, all the computations have been recalculated, including solvent at (SMD) ω -B97xD/6-31G (d,p). Neither the cavity nor the geometry of the newly calculated species differs substantially from the calculated ones on the original submission.

7. In the experimental section on page 4 of the SI, the ^{31}P NMR data for compounds 5 and 6 appear to be swapped. They don't match the spectra reported later on in the SI or the data quoted in the main text.

Response: The ^{31}P NMR data for compounds **5** and **6** have been placed in the correct locations.

8. In the experimental section, the MS results are probably better reported as "[M+H]⁺", not "[M+1]⁺".

Response: All the "[M+H]⁺", have been changed to "[M+1]⁺".

We hope that these amends address the reviewers' concerns. The authors appreciate your time for considering our manuscript and look forward to your positive response.

Looking forward to hearing back from you.

Yours Sincerely,

Felipe García

REVIEWERS' COMMENTS:

Reviewer #1 (Remarks to the Author):

The corrections that have been made by authors are found to be sufficient. The paper is acceptable for publishing in Communications Chemistry.

Reviewer #2 (Remarks to the Author):

The authors have done a thorough job at addressing the questions raised in the revised MS and SI, and I'm satisfied with their response on the technical part of the paper and support their publication.

However, I do feel that the increased discussion on the calculated host-guest properties did not make the paper stronger considering none can be experimentally verified. It distracts readers from the solid discussions on their rather neat synthetic strategy (the most important contribution of this paper).

I suggest that the limitations of the experimental host-guest properties should be pointed out earlier on, before the discussion of the computational studies. The latter could also be shortened and possibly moved in the supporting information.

Reviewer #3 (Remarks to the Author):

The authors have thoughtfully considered all of the reviewer comments and did an impressive amount of work to address them. I recommend publication of the manuscript in its current form.

Rebuttal Letter

Dear all,

First of all, I would like to thank you for the time and effort taken to revise our manuscript "**Size-control in the synthesis of oxo-bridged phosphazane macrocycles via a modular addition approach**" by Xiaoyan Shi, Felix León, How Chee Ong, Rakesh Ganguly, Jesús Díaz, and myself" which we sent for your consideration for publication in *Communications Chemistry (NPG)*.

We have addressed point-by-point the issues raised by the reviewers

REVIEWERS' COMMENTS:

Reviewer #1 (Remarks to the Author):

The corrections that have been made by authors are found to be sufficient. The paper is acceptable for publishing in *Communications Chemistry*.

ANSWER: We thank Reviewer for the time he time and effort taken to revise our manuscript and supporting the publication of the manuscript.

Reviewer #2 (Remarks to the Author):

The authors have done a thorough job at addressing the questions raised in the revised MS and SI, and I'm satisfied with their response on the technical part of the paper and support their publication.

However, I do feel that the increased discussion on the calculated host-guest properties did not make the paper stronger considering none can be experimentally verified. It distracts readers from the solid discussions on their rather neat synthetic strategy (the most important contribution of this paper).

I suggest that the limitations of the experimental host-guest properties should be pointed out earlier on, before the discussion of the computational studies. The latter could also be shortened and possibly moved in the supporting information.

ANSWER: We thank Reviewer 2 for the comments and the suggestion. Firstly, we have highlighted the limitation on assessing the experimental host-guest properties of 7 and placed it before the host-guest theoretical studies (highlighted in green). Secondly, we have shortened the host-guest theoretical discussion and moved most of it to the to the SI (highlighted in blue).

Reviewer #3 (Remarks to the Author):

The authors have thoughtfully considered all of the reviewer comments and did an impressive amount of work to address them. I recommend publication of the manuscript in its current form.

ANSWER: We thank Reviewer for the time he time and effort taken to revise our manuscript and supporting the publication of the manuscript.

We hope that these amends address the editorial concerns. The authors appreciate your time for considering our manuscript and look forward to your positive response.

Looking forward to hearing back from you.

Yours Sincerely,

Felipe García